# Circulating human anti nucleolus antibody (ANCAb) and biochemical parameters in type 2 diabetic patients with and without complications

**Heevi Ameen Rajab[1], Alan Bapeer Hassan[2], Israa Issa Hassan[2], Deldar Morad Abdulah[3], Farsat Saeed Saadi[4]** *

1 Medical Chemistry Department, College of Medicine, University of Duhok-Iraq, Duhok, Iraq, 2 Basic Sciences Unit, College of Nursing, University of Duhok-Iraq, Duhok, Iraq, 3 Community Health Unit, College of Nursing, University of Duhok-Iraq, Duhok, Iraq, 4 Head of CPD Department, Duhok General Directorate of Health-Iraq, Duhok, Iraq

* farsat.saeed.saadi@gmail.com

**Data Availability Statement:** All relevant data are within the manuscript and Supporting Information files.

## Abstract

### Introduction

There is no evidence on the role of Human Anti Nucleolus Antibody (ANCAb) in type 2 diabetes mellitus (T2DM). We compared prevalence and concentration of ANCAb between age and a gender-matched sample of T2DM with and without diabetes-related complications.

### Methods

In this study, the reaction to ANCAb was compared quantitatively between 38 T2DM patients complicated with microvascular conditions and 43 T2DM without complications as controls.

### Results

The patients in complicated and non-complicated groups were comparable in diabetes duration (9.0 vs. 5.0 years; P = 0.065), respectively. The study found that 27 cases (71.1%) of the complicated group reacted to ANCAb test compared to 25 (58.1%) in non-complicated patients (P = 0.226; 3.53 vs. 2.72 ng/mL; P = 0.413). The reaction response to ANCAb in patients with neuropathy and cardiovascular complications was 80.0%, 76.2% in patients with neuropathy compared to 58.1% in the control group (P = 0.398). The reaction response to ANCAb in patients with mono-complication was 72.7% compared 68.8% in patients with multi-complication (P = 0.466). Similarly, 76.2% of patients with T2DM and complicated with neuropathy (n = 21 patients) reacted to ANCAb compared to 58.1% in control patients with (P = 0.158).

### Conclusions

Reaction to ANCAb was not statistically different between the T2DM patients with and without complications.

**Funding:** The study was not funded by any organization.

**Competing interests:** The authors do not declare any conflict of interest.

## Introduction

Diabetes is a rising social and epidemiological issue. Patients with type 2 diabetes mellitus (T2DM) are at risk of frequent complications, included cardiovascular, renal, neural, and retinal diseases. In addition, T2DM has significant morbidity and mortality across the world [1]. For instance, population-based studies have reported that majority of T2DM patients develop diabetic peripheral neuropathy (DPN), and close to 30% of these patients have clinical manifestations [2, 3]. Importantly, up to 50% of patients with DPN are asymptomatic [4]. Exact factors contributing to development of T2DM are unclear, but it seems that diabetes has more diverse etiology, than previously thought.

The studies conducted in type 1 diabetes mellitus (T1DM) patients with severe autonomic neuropathy have documented that lymphocytic infiltration and small nerve fiber damage in autonomic ganglia indicates a vigorous immune response [5]. Circulating auto-antibodies in sera of patients with T1DM have been reported that make a reaction with autonomic conditions, such as sympathetic ganglia and vagus nerve. This reaction might be associated with the future development of autonomic neuropathy [6].

About DPN, Granberg, Ejskjaer [6] mentioned that it is unclear whether autoimmunity has a primary role in the disease pathogenesis, or it escalates the DPN initiation by metabolic or vascular injury.

The anti-GAD65 (glutamic acid decarboxylase) antibody is a strong predictive marker for the onset of T1DM [7]. The presence of this antibody in patients with recent-onset T1DM was associated with hyperglycemia and reduced peripheral nerve function, suggesting a common mechanism for β-cell and neuronal damage [8]. Decreased motor nerve conduction velocities in the median, ulnar and peroneal nerves were observed in patients who have high GAD65 antibodies along with prolonged F wave latencies, high thermal threshold detection for hot and cold, and decreased cardiovascular autonomic functions [8].

A study conducted by Srinivasan, Stevens [9] demonstrated that serum of T2DM patients with neuropathy contains an autoimmune immunoglobulin that induces independent complement and calcium-dependent apoptosis in neuronal cells. An intense fluorescence on the surface of neuronal cells has been shown to relate to anti-human IgG antibody in neuropathic T2DM patients [9]. Some other inflammatory biomarkers have been seen in patients with coronary artery diseases (CAD) [6]. Unsuitable inflammatory responses maybe related to both increased chronic development of atherosclerotic plaques and an increased plague rupture associated with acute myocardial infarction [10].

Although different studies suggested that antibodies may be present in the sera of T2DM patients with complications, the development of complications in these patients needs further investigation. There is no definitive report of the prevalence of a systemic immune reaction, characterized by the presence of high titers of antinuclear antibodies (ANA) in T2DM patients with chronic microvascular complications yet [11–13].

The rate of autoimmune diseases in the general population is rising more than it previously reported in the literature [14, 15]. In addition, it has been reported that patients with T2DM have more occurrence of autoimmune diseases that it was previously reported [1]. An increased destruction of cells in various organs following microangiopathy and macroangiopathy may induce a secondary immune response. This response may aggravate further course of chronic complications. Anti-nuclear antibodies (ANA) could be a biomarker or be directly involved in chronic complications of diabetes since it is produced in response to cell necrosis or cell apoptosis. Therefore, this kind of relation may be a notion for the higher prevalence of antinuclear antibodies in more severe forms of coronary atherosclerosis [1].

The role of reaction to ANA in T2DM patients with complications has been investigated in a few studies, mostly by a qualitative and in a few investigations by a semi-quantitative method. Given to controversy findings on the role of ANA in the development of the complications in T2DM [11, 12]. But, the development of micro-complications in patients with T2DM could be searched in an anti- nucleolus level rather than nuclear one. Human Anti Nucleolus Antibody (ANCAb) may have a role in the development of complications. With the currently available hints on the possible role of ANA in the development or escalation of T2DM, it is so valuable to examine this role in a sample of complicated and non-complicated patients. The prevalence of reaction to ANCAb was compared between a sample of patients with T2DM and complications and a sample of T2DM without complications. The investigators anticipated the higher prevalence of reaction to ANCAb in T2DM patients with complications compared to those patients without complications.

## Patients and methods

### Study design and sampling

The patients who were diagnosed with T2DM by the study internist were screened physically and clinically for the eligibility criteria of the study. In the current case-control study, 38 patients with T2DM (41–64 years old) affected by various complications, including neuropathy, retinopathy, cardiovascular, or nephropathy were included and their biochemical parameters and ANCAb reaction were compared with 43 non-complicated patients with T2DM (37–64 years old). The T2DM patients in both study groups were matched in age (P = 0.115) and gender (P = 0.244) prior to study analysis. The complicated patients with neuropathy, retinopathy, cardiovascular, or nephropathy were considered exposed/case group. The patients who attended the out-patient clinic for medical check-up or receiving their regular medications were screened physically and clinically by the study internist. The study internist has weekly regular clinical duty in the out-patient clinic. The patients were screened for the possible complications of T2DM and consisted microvascular diseases, neuropathy, nephropathy, and retinopathy. The patients who were diagnosed to have one of the mentioned above complications following diabetes were categorized as T2DM patients with complication. The T2DM patients who did not have the complications were categorized as T2DM patients without complications.

The patients were recruited purposively from Duhok Diabetes Center of Azadi Teaching Hospital following taking ethical approval from the local department, and written consent forms from patients. The data collection was performed between 3rd November 2018 and 24th February 2019.

The patients who attended the center for medical check-up or treatment were screened consecutively for the inclusion criteria. The internist of the study (fifth author) performed physical and clinical examinations for all patients for medical conditions and diabetes complications before inclusion in the study.

### Inclusion and exclusion criteria

The patients who were diagnosed with T2DM with and without microvascular complications aged 35 years and older of both gender and have diabetes since the last five years regardless of their socio-demographic aspects were eligible to be recruited in this study. Those with autoimmune disease including either organ-specific illnesses (e.g., thyroid disease, type 1 diabetes mellitus (T1DM), and myasthenia gravis) or systemic diseases (e.g., rheumatoid arthritis (RA), systemic lupus erythematosus (SLE), and pregnant patients, active infection or inflammation, and the patients on hormone replacement therapy were excluded through the clinical examination and self-reported technique.

The patients who were included in this study were previously diagnosed with T2DM. The patients were consecutively screened through physical and clinical examinations and history taking for eligibility criteria. The information such as age, gender, disease duration, and anthropometric measures were recorded in a pre-designed questionnaire. The treatments of the patients were checked by the study internist. The internist asked the patients about the history of disease and its complications to be sure that the current complications belong to the disease. To include the suitable sample in the study, the patients were screened for the disease comorbidities.

### Diagnosis and measurement criteria

The diagnosis of T2DM was established in accordance with the criteria of the World Health Organization [16]. The diagnosis of the medical conditions was established through the medical and clinical examinations by the study internist. In this investigation, cardiovascular disease was defined as a class of diseases affecting the heart and blood vessels, as explained by the WHO [17]. The patients who had the following 12 cardiovascular presentations following diabetes development were considered the cases with CVD complications. The CVD conditions were: stable angina, unstable angina, heart failure, unheralded coronary death, myocardial infarction, ischemic stroke, transient ischemic attack, peripheral arterial disease, subarachnoid hemorrhage, intracerebral hemorrhage, abdominal aortic aneurysm and a composite outcome consisted of cardioversion, ventricular arrhythmia, implantable cardioverter defibrillator, and cardiac arrest [18]. The HbA1C <7% was considered controlled diabetes and ≥7% as uncontrolled diabetes [19].

### Diabetic peripheral neuropathy (DPN) diagnosis

Polyneuropathy-neurological examination was performed for superficial sensation, by indicators such as pain, touch, and temperature. Also, the following tests were performed: vibration perception with tuning fork and monofilament test. The neuropathy was diagnosed in accordance with the criteria of the American Academy of Neurology (AAN) through a combination of neuropathic symptoms, multiple signs, and abnormal electro-diagnostic studies [20].

### Nephropathy

The diabetic nephropathy was diagnosed according to the clinical examination and biochemical parameters by the study internist [21]. The albumin was measured in a spot urine sample as the first step in the screening and diagnosis of diabetic nephropathy during medical visit as recommended by American Diabetes Association guidelines [22]. The following albuminuria cutoff were considered for microalbuminuria and macro-albuminuria stages [23].

### Retinopathy

The patients who had a complaint on their visions were referred to an ophthalmologist for the vision tests for the disease confirmation. The patients who had the following symptoms were screened for diabetic retinopathy. The symptoms for the diagnosis of diabetic retinopathy were finding spots or floaters, blurred vision, having a dark or empty spot in the center of your vision, and difficulty seeing well at night. Accordingly, the diagnostic clinical manifestations were leaking blood vessels, swelling inside the retina, pale, fatty deposits on the retina, damaged nerve tissue, and any changes to the blood vessels [24].

## Biochemical measurements

The investigators collected 10 ml of the venous blood samples from all patients following overnight fasting (= >12 hrs.) by an experienced phlebotomist. The obtained samples were centrifuged at 3,000 rpm at 4°C for 15 min. Their supernatants were decanted and frozen at -30°C and assayed. The biochemical measurements were determined using Roche autoanalyzer 6000 Cobas (Roche Diagnostics, Mannheim, Germany) in the medical lab of the hospital. In this study, the biochemical measurements were calculated from the serum samples. The biochemical parameters, including, lipid profile, liver function tests, renal function tests, insulin, and HbA1C, were measured through the biochemical techniques. The biochemical measurements were performed in the medical lab of Azadi Teaching Hospital in Duhok city.

## Anthropometric measurements

The patients' weight was measured in Kg scaled by Bathroom Scale, a digital scale nearest to 0.1 cm. Waist circumference (WC) was measured in the horizontal plane by non-stretch tape in the midway place between the lateral lower ribs and the iliac crests by Double-Scale Soft Tape with the nearest 0.1 cm. Recommended cut-offs for increased health risk are a waist circumference of more than 102 cm for men and more than 88 cm for women [25].

## Human anti nucleolus antibody measurement

The measurement of serum ANCAb was performed using Human Anti Nucleolus Antibody (ANCA, (Competitive ELISA Kit), Catalog Number: MBS7228076–96 tests from MyBioSource, Inc. The USA. ELx800 Universal Microplate Reader with ELx50 Auto Strip Washer was used for ANCAb measurement. This ANCAb ELISA kit is a 1.5-hour solid-phase ELISA designed for the quantitative determination of Human ANCAb. The concentration of antibody was determined by comparison to a standard curve generated by known concentrations of ANCAb ranged 0–100 ng/mL [26].

In the current study, any reaction of ANCAb with components of the body's healthy cells was considered a reacted ANCAb. ANCAb specifically target substances originated in the nucleolus of a cell, hence the name "antinucleoli" They probably do not damage living cells because they cannot access their nucleoli. However, ANCAb can cause damage to tissue by reacting with nuclear substances when they are released from injured or dying cells [26].

ANCAb ELISA kit applies the competitive enzyme immunoassay technique. This technique utilizes Nucleolus antigen and an ANCAb-HRP conjugate. The assay sample and buffer were incubated together with ANCAb-HRP conjugate in pre-coated plate for one hour. Following the incubation period, the wells were decanted and washed five times and the wells were incubated with a substrate for HRP enzyme. The product of the enzyme-substrate reaction was a blue colored complex. Finally, a stop solution was added to stop the reaction. Subsequently, the color turned to the yellow. The color intensity was measured spectrophotometrically at 450 nm in a microplate reader. The intensity of the color was inversely proportional to the ANCAb concentration, because ANCAb from samples and ANCAb-HRP conjugate compete for the Nucleolus antigen binding site. Since the number of sites is limited, as more sites are occupied by ANCAb from the sample, fewer sites were left to bind ANCAb-HRP conjugate. A standard curve was plotted relating the intensity of the color (O.D.) to the concentration of standards. The ANCAb concentration in each sample was interpolated from this standard curve.

The sensitivity of this assay was 1.0 ng/mL. This assay has high sensitivity and excellent specificity to detect ANCAb. There is no significant cross-reactivity or interference between ANCAb and analogues. **NOTE**: The catalogue mentioned that the assay is limited by current

skills and knowledge. It added that it was impossible to complete the cross-reactivity detection between ANCAb and all the analogues, therefore, cross reaction may still exist in some cases.

## Statistical analysis

The normality of the biochemical and general parameters between the patients in both complicated and non-complicated patients was tested through drawing a histogram between two groups. The outliers of the biochemical parameters in both study groups were examined and deleted by a Box-Plots. The frequency percentage and mean standard deviation was performed for descriptive purposes of the study. The comparison of general characteristics between the patients in both study groups was examined in an independent t-test, Pearson Chi-square test, Fishers' exact test, or Mann-Whitney U-test.

The comparison of biochemical indicators between the patients in complicated and non-complicated groups was examined in an independent t-test. The difference in the prevalence of ANCAb reaction between the complicated and non-complicated patients was examined in Pearson chi-squared or Fishers' exact tests. The level of ANCAb was transformed trough SQRT method, and the comparison of the median values of the complicated and non-complicated was examined in Mann-Whitney U-test. The magnitude of the association was calculated in an odds ratio. The level of less than 0.05 was considered the statically significant difference. The SPSS version 24 is used for the statistical calculations.

## Ethical approval

The ethical approval of the present investigation was obtained from the Scientific Research Division, Directorate of Planning, Duhok Directorate General of Health, registered as reference number 17042018–3 on 28[th] October 2018 in accordance with the principles of the Declaration of Helsinki [27].

## Results

The study revealed that the patients with and without diabetic complication were comparable in age (53.34 vs. 50.90 years, P = 0.115), and diabetes duration (9.0 vs. 5.0 years; P = 0.065), number of patients with high school certificate and higher (5.3% vs. 11.6%; P = 0.439), smokers (13.2% vs. 14.0%; P = 0.917), and gender (Male: 31.6% vs. 44.2%; P = 0.244), respectively. The patients in both with and without diabetic complication were similar in BMI (27.92 vs. 29.26; P = 0.175), while the patients without diabetic complication had a greater waist circumference (107.95 cm) compared to patients with diabetic complication (102.30 cm; P = 0.021), as presented in Table 1.

The study showed that the patients with and without complication were comparable in most biochemical parameters (P>0.05). However, patients with diabetic complication had a lower concentration of creatinine (0.68 vs. 0.77 mg/dL; P = 0.041) and urea (27.78 vs. 33.12 mg/dL; P = 0.013), respectively compared to the patients without diabetic complication (Table 2).

The study showed that 27 cases (71.1%) in patients with diabetic complication reacted to the ANCAb test compared to 25 (58.1%) in patients without diabetic complication, while the overall difference of prevalence was not statistically significant (P = 0.226). The median values of the transformed ANCAb in both groups of patients with and without diabetic complication were 3.53 and 2.72 ng/mL, respectively with no statistically significant difference (P = 0.413), as presented in Table 3 and Fig 1A.

The study did not show the significant difference in the concentration of ANCAb between the patients who had controlled and uncontrolled T2DM in patients with and without diabetic complication (Table 4 & Fig 1D).

**Table 1. Comparison of general information between type 2 diabetes mellitus with and without complications.**

| Patients' characteristics | Patient with diabetic complication (n = 38) | Patient without diabetic complication (n = 43) | P-Value (two-sided) |
|---|---|---|---|
| **Age (year)** | 53.34 (6.94) | 50.93 (6.63) | 0.115[a] |
| **Range: 37–64 years** | 41–64 | 37–64 | |
| **Waist Circumference (cm) Range: 80–126 cm** | 102.30 (11.97) 80–125 | 107.95 (8.65) 92–126 | 0.021[a] |
| **WC Categories** | | | 0.275[b] |
| Healthy | 13 (34.2) | 10 (23.3) | |
| Unhealthy | 25 (65.8) | 33 (76.7) | |
| **Diabetes duration; Median/Interquartile Range Range: 0.5–30 years** | 9.0 (6.0) | 5.0 (7.0) | 0.065[d] |
| **BMI** | 27.92 (4.45) | 29.26 (3.91) | 0.175[a] |
| Range: 17.58–38.31 | 17.58–34.79 | 20.14–38.31 | |
| **Education Categories** | | | 0.439[c] |
| Under High School | 36 (94.7) | 38 (88.4) | |
| High School and Higher | 2 (5.3) | 5 (11.6) | |
| Smokers | 5 (13.2) | 6 (14.0) | 0.917[b] |
| **Gender** | | | 0.244[b] |
| Male | 12 (31.6) | 19 (44.2) | |
| Female | 26 (68.4) | 24 (55.8) | |

The following statistical tests were performed:

[a]Independent t-test

[b]Pearson Chi-square test

[c]Fishers' exact test

[d]Mann-Whitney U-test

To have a better picture of the reaction to ANCAb in patients with different types of complications, the prevalence of ANCAb reaction in different categories of complications was presented in Table 5 & Fig 1C. The study revealed that the patients with neuropathy along with cardiovascular complications had a higher reaction response to ANCAb (80.0%) followed by those with neuropathy (76.2%) compared to 58.1% in the non-complicated group (P = 0.398). Similarly, the patients with mono-complication had a higher reaction response to ANCAb (72.7%) compared to those patients with multi-complication (68.8%), but the overall difference in prevalence was not statistically significant (P = 0.466), as presented in Table 5 & Fig 1B.

The same comparison was performed between the patients with neuropathy and non-complicated in Table 5. The examination showed that 76.2% of patients with T2DM and complicated with only neuropathy (n = 21 patients) reacted to ANCAb compared to 58.1% in non-complicated patients with no statistically significant difference (P = 0.158).

## Discussion

The present study showed that there is a difference in the number of the cases reacted to ANCAb test in the complicated patients compared to non-complicated patients, while the overall difference in reaction prevalence was not statistically significant either among multi-complication, mono-complication, and non-complicated patients or between the T2DM patients with neuropathy and non-complicated patients. Besides, there was no substantial difference in the ANCAb concentration between complicated and non-complicated patients in either category.

**Table 2. Comparison of biochemical parameters between type 2 diabetic patients with and without complications.**

| Biochemical Parameters | Study Groups | | |
|---|---|---|---|
| | Patient with diabetic complication (n = 38) | Patient without diabetic complication (n = 43) | P-Value (two-sided) |
| Total cholesterol; mg/dL | 188.89 (41.88) | 189.19 (41.20) | 0.975[a] |
| HDL; mg/dL | 44.89 (11.02) | 40.40 (9.26) | 0.051 [a] |
| LDL; mg/dL | 105.86 (34.03) | 112.39 (30.82) | 0.387 [a] |
| T.G; mg/dL | 157.21 (56.21) | 171.67 (66.09) | 0.319 [a] |
| Albumin; g/dL | 4.43 (0.31) | 4.54 (0.29) | 0.121 [a] |
| Urea; mg/dL | 27.78 (8.13) | 33.12 (10.59) | **0.013** [a] |
| Creatinine; mg/dL | 0.68 (0.18) | 0.77 (0.20) | **0.041** [a] |
| U.A; mg/dL | 4.55 (1.34) | 4.50 (1.08) | 0.838 [a] |
| GOT; U/L | 16.11 (4.82) | 17.88 (4.07) | 0.089 [a] |
| GPT; U/L | 19.84 (6.38) | 22.37 (8.43) | 0.136 [a] |
| Alkaline Phosphatase; U/L | 96.21 (29.67) | 105.84 (37.29) | 0.211 [a] |
| HbA1C; % | 8.30 (1.96) | 7.79 (1.19) | 0.171 [a] |
| **Range: 4.60–13.20** | 4.60–13.20 | 5.90–10.0 | |
| **HbA1C Categories** | | | 0.696[b] |
| Uncontrolled Diabetes | 28 (73.7) | 30 (69.8) | |
| Controlled Diabetes | 10 (26.3) | 13 (30.2) | |

[a]The independent t-test and

[b] Pearson Chi-squared test were performed for statistical analyses.

The bold numbers show a significant difference.

The presence of ANA has been shown to associate with an autoimmune disease in a few investigations. For example, Grainger and Bethell [12] determined the role of the presence of high titers of antinuclear antibodies in the development of coronary atherosclerosis. They measured the serum ANA of the 40 patients aged 53–76 years old with at least 50% stenosis of

**Table 3. Comparison of human anti-nucleolus antibody between type 2 diabetic patients with and without complications.**

| Human Anti Nucleolus Antibody (ANCAb) | Study Groups | | OR (95% CI) | P-Value (two-sided) |
|---|---|---|---|---|
| | Patient with diabetic complication (n = 38) | Patient without diabetic complication (n = 43) | | |
| **ANCAb reaction** | | | | 0.226[a] |
| Reacted | 27 (71.1) | 25 (58.1) | 1.77 (0.7–4.46) | |
| Non-reacted | 11 (28.9) | 18 (41.9) | | |
| **Males** | | | 1.80 (0.40–8.07) | 0.440[a] |
| **Reacted** | 8 (66.7%) | 10 (52.6%) | | |
| **Non-Reacted** | 4 (33.3%) | 9 (47.4%) | | |
| **Females** | | | 1.63 (0.49–5.39) | 0.423[a] |
| **Reacted** | 19 (73.1%) | 15 (62.5%) | | |
| **Non-Reacted** | 7 (26.9%) | 9 (37.5%) | | |
| ANCAb, ng/mL | 3.53 (4.91) | 2.72 (4.83) | | 0.413[b] |
| Range: 0.0–10.87 | 0.0–10.87 | 0.0–9.29 | | |
| **Males** | 3.37 (4.86) | 2.54 (4.55) | | 0.768 |
| **Females** | 3.88 (4.91) | 2.75 (4.98) | | 0.540 |

[a]Pearson Chi-squared and

[b]Mann Whitney U-test was performed for statistical analyses.

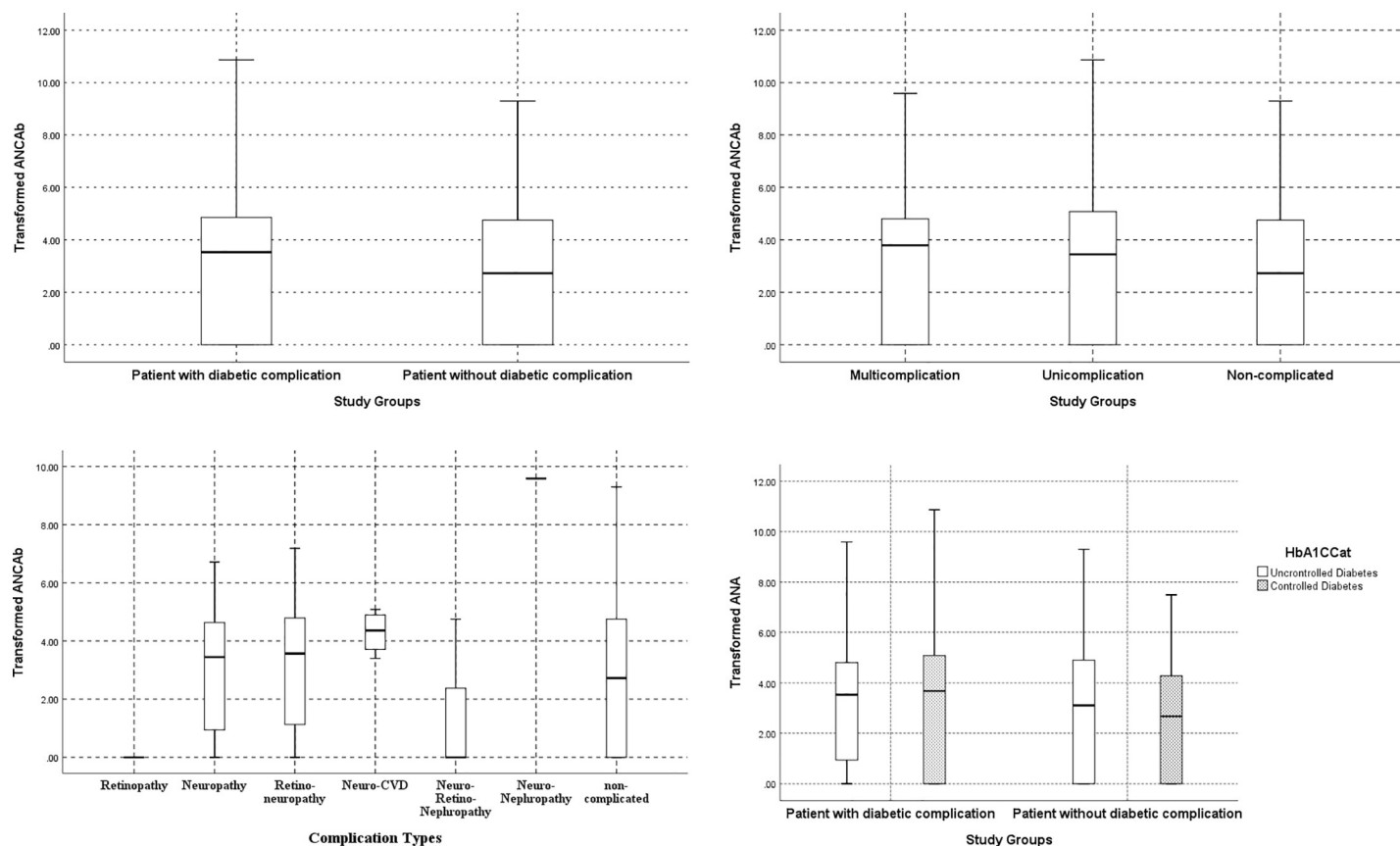

**Fig 1.** a. Presentation of ANCAb concentration (transformed ANCAb) between patients with and without diabetic microvascular complications. b. Presentation of ANCAb (transformed ANCAb) among patients with uni and multi microvascular complications. c. Presentation of ANCAb (transformed ANCAb) among patients with different microvascular complications. d. Presentation of ANCAb (transformed ANCAb) among study groups based the HbA1C control.

three main coronary arteries and compared ANA concentration with it in 30 patients aged 48–74 years old with no evidence of coronary atherosclerosis. The ANA presented by immunofluorescent detection of human antibodies bound to HEp-2000 cells was detected at least in 1/40 in 28 (70%) of the cases, but only five (17%) of the controls (OR: 11.67; 95% CI: 3.91–17.82; P<0.001). The cases with severe coronary atherosclerosis had higher titers of ANA compared to those patients with normal coronary arteries. The similar findings were reported in T2DM and T1DM patients affected with peripheral neuropathy. Janahi, Santos [11] included 30 patients with DPN, 30 diabetic control patients without neuropathy, and 20 healthy control in a case-control study. The investigators found the significant positive reaction of ANA in the serum of the DPN compared to the control groups with the odds 50 times higher of positive values of ANA in the neuropathy group compared to the control groups. Besides, a significant

**Table 4. Comparison of ANCAb between controlled and uncontrolled diabetes in study groups.**

| Study Groups | HbA1C Categories | | P-Value (two-sided) |
|---|---|---|---|
| | Uncontrolled Diabetes | Controlled Diabetes | |
| Patient with diabetic complication | 3.53 (4.36) | 3.67 (5.27) | 0.893 |
| Patient without diabetic complication | 3.11 (4.97) | 2.67 (4.55) | 0.690 |

Mann-Whitney U-test was performed for statistical analyses.

**Table 5. Association of ANCAb reaction to complications categories in patients with T2DM.**

| Compilations | Total Complications | | Reaction to ANCAb | | P-Value (two-sided) |
|---|---|---|---|---|---|
| | No (%) | Mean (SD); ng/mL | Reacted | Non-reacted | |
| **Complication Categories** | | | | | 0.170[c] |
| Retinopathy | 1 (1.2) | 0.0 | 0 (0.0) | 1 (100.0) | 0.398[b] |
| Neuropathy | 21 (25.9) | 3.43 (2.69) | 16 (76.2) | 5 (23.8) | |
| Retino-neuropathy | 7 (8.6) | 3.23 (2.66) | 5 (71.4) | 2 (28.6) | |
| Neuro-CVD | 5 (6.2) | 3.44 (2.03) | 4 (80.0) | 1 (20.0) | |
| Neuro-Retino-Nephropathy | 3 (3.7) | 1.58 (2.74) | 1 (33.3) | 2 (66.7) | |
| Neuro-Nephropathy | 1 (1.2) | 9.59 | 1 (100.0) | 0 (0.0) | |
| non-complicated | 43 (53.1) | 2.75 (2.69) | 25 (58.1) | 18 (41.9) | |
| **Complication Numbers** | | | | | 0.643[c] |
| Multi-complication | 16 (19.8) | 3.38 (2.86) | 11 (68.8) | 5 (31.3) | 0.466[a] |
| Mono-complication | 22 (27.2) | 3.27 (2.73) | 16 (72.7) | 6 (27.3) | |
| Non-complicated | 43 (53.1) | 2.75 (2.69) | 25 (58.1) | 18 (41.9) | |
| **Complication Categories^** | | | | | 0.350[c] |
| Neuropathy | 21 (32.8) | 3.43 (2.69) | 16 (76.2) | 5 (23.8) | 0.158[a] |
| non-complicated | 43 (67.2) | 2.75 (2.69) | 25 (58.1) | 18 (41.9) | |

[a]Pearson Chi-squared

[b]Fishers' exact, and

[c]ANOVA one-way tests were performed for statistical analyses.

^Phi test: Value 0.177 (Phi tests shows a positive non-significant association)

The first P-value is for the mean difference and the second one is for the prevalence difference.

correlation was found between the presence of ANA and the neurological manifestation of neuropathy. The neurological manifestations were neuropathy symptom score, neuropathy disability score, and vibration perception threshold.

We examined this kind of association with different complications in a quantitative technique in patients with T2DM; however, we did not find the significant difference in the prevalence of positive ANCAb between patients in the complicated and non-complicated groups despite the prevalence of reaction was high in our cases (71.1%). It is important to mention that Janahi, Santos [11] included both T1DM and T2DM in DPN and diabetic groups. Importantly, the etiology of these two types of diabetes are different since T1DM is an autoimmune disease. Even there is an early appearance of anti- GAD65– specific T-cells in T1DM, and it has a strong role in the onset of the T1DM disease [7]. It has been reported that T1DM patients with high GAD65 antibodies have a positive correlation with neurological defects [8].

Zinman, Kahn [28] found a similar β-cell function between GAD-positive and -negative patients. However, GAD positive patients had higher HDL cholesterol and lower triglyceride levels. Hathout, Thomas [29] reported 8.1% of positive ICAs, 30.3% of positive GADs, and 34.8% of positive IAAs in a pediatric population with T2DM. We refer the high titers of reaction to ANCAb in cases and control in this investigation (71.1% and 58.1%, respectively) to latent autoimmune diabetes in adults (LADA) [30]. It claims that autoimmunity is the key responsible factor of LADA, because diabetes has biochemical markers of β-cells directed autoimmunity [30]. Moreover, LADA is related to a "mild" form of T1DM. However, it is not clear, whether the ANA detected in the neuropathy sample relates to DM itself or antibodies related to neuropathy.

It is interesting to mention that most of the currently available investigations have performed the antinuclear antibodies in a qualitative way. Therefore, patients with little

inflammation have responded positively to the test. Even the patients with LADA could have a positive reaction to the autoimmunity test. The findings of Grainger and Bethell [12] in their second step, which was conducted in a semi-quantitative method approve our results.

As Grainger and Bethell [12] mentioned, this association might infer either a correlative association or a causative association in either direction. Also, the implication of a pro-inflammatory phenotype in the development of vascular diseases might lead to some mis-regulation of inflammatory responses and independently results in increased development of atherosclerosis and increased incidence of ANA positivity or vice versa. Repeated myocardial necrosis, even on a little scale, could be adequate to result in leakage of intracellular antigens and induces ANA production. They hypothesized that the presence of ANA might potentiate the development of atherosclerosis through helping the local inflammatory response at the site of lipid deposition into the vascular wall where cellular necrosis occurs.

We found that non-complicated group has significantly higher value in waist circumference and a trend to higher BMI compared to complicated group. This finding confirms the role of adipose tissues (AT) in systemic and B cell intrinsic inflammation, decreased B cell responses and autoimmune antibodies' secretion. The adipocytes in the human obese subcutaneous AT (SAT) enable to secrete several pro-inflammatory cytokines and chemokines that are responsible to the establishment and maintenance of local systemic inflammation. Obesity is responsible for decreased oxygen and consequent hypoxia and cell death result in further secretion of pro-inflammatory cytokines, "self-protein antigens, cell-free DNA and lipids [31–33].

The currently available review study explained the assumption of the pathogenesis of T2DM that encompasses autoimmune aspects and it is recognized increasingly based on the presence of circulating autoantibodies against β cells, self-reactive T cells, and on the glucose-lowering efficacy of some immunomodulatory therapies in T2DM. The LADA in elderly patients with the slowly progressive manifestation of diabetes resulted in a combination of T2DM and T1DM features. Although the autoantibody cluster of the patients with LADA and T1DM are different, the presence of these indicators reflects steady progression toward β-cell death and subsequent requirement for initiation of insulin treatment in a shorter period compared to autoantibody-negative T2DM patients. The autoimmune activation in T2DM appears to be contributed to the chronic inflammatory status. Cryptic 'self' antigens induce an autoimmune response in the case of inflammation-induced tissue destruction that accelerates β-cell destruction [13]. We recommend further quantitative investigations on the comparison of ANA/ANCAb between the T2DM with and without microvascular complications.

## Strengths and limitations

This is the sole study that determined the ANCAb reaction in a quantitative technique. We applied restricted inclusion and matching criteria to include the patients in both complicated and non-complicated groups. However, the findings reported in this study must be interpreted with caution since the quantitative technique was used to measure the ANCAb in contrast with the literature. It is hard to make a between-study comparison due to using different titers to measure ANA since the patients with high titers ($\geq$1:640) are more susceptible to autoimmune disease compared to the patients with low ANA titers ($\leq$1:640) [34]. It was hard for the investigators to include a large sample size in this study owing to financial restriction and patients' availability and we did not include the healthy control group. In addition, we did not verified the results with another measurement technique owing to financial resources. Besides, we did not perform the ocular tests for all patients. The patients who had vision complain during physical and clinical investigations were referred to an ophthalmologist only.

## Conclusions

This study found that the concentration and reaction of the ANCAb to autoimmune antibodies was not statistically different between the T2DM patients with and without complications, either in uni-complication or multi-complications, or types of microvascular complications. In addition, the ANCAb concentration was not statistically different in patients with controlled or uncontrolled T2DM. The authors suggest to examine the same autoimmune antibody in one type of microvascular complication and with a larger sample.

## Supporting information

**S1 File.**
(PDF)

**S2 File.**
(SAV)

## Author Contributions

**Conceptualization:** Heevi Ameen Rajab, Alan Bapeer Hassan, Israa Issa Hassan, Deldar Morad Abdulah, Farsat Saeed Saadi.

**Data curation:** Heevi Ameen Rajab, Alan Bapeer Hassan, Israa Issa Hassan, Deldar Morad Abdulah, Farsat Saeed Saadi.

**Formal analysis:** Alan Bapeer Hassan, Israa Issa Hassan, Deldar Morad Abdulah, Farsat Saeed Saadi.

**Funding acquisition:** Heevi Ameen Rajab, Alan Bapeer Hassan, Deldar Morad Abdulah, Farsat Saeed Saadi.

**Investigation:** Heevi Ameen Rajab, Alan Bapeer Hassan, Israa Issa Hassan, Deldar Morad Abdulah, Farsat Saeed Saadi.

**Methodology:** Alan Bapeer Hassan, Israa Issa Hassan, Deldar Morad Abdulah, Farsat Saeed Saadi.

**Project administration:** Deldar Morad Abdulah.

**Resources:** Heevi Ameen Rajab, Alan Bapeer Hassan.

**Software:** Deldar Morad Abdulah.

**Supervision:** Deldar Morad Abdulah.

**Writing – original draft:** Alan Bapeer Hassan, Deldar Morad Abdulah.

**Writing – review & editing:** Heevi Ameen Rajab, Alan Bapeer Hassan, Israa Issa Hassan, Deldar Morad Abdulah, Farsat Saeed Saadi.

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
