## [Decision Letter · Decision Letter 0]

14 May 2020

PONE-D-20-08208

Circulating Human Anti Nucleolus Antibody (ANCAb) and Biochemical Parameters In Type 2 Diabetic Patients with and without Complications

PLOS ONE

Dear Mr. Abdulah,

Thank you for submitting your manuscript to PLOS ONE. After careful consideration, we feel that it has merit but does not fully meet PLOS ONE’s publication criteria as it currently stands. Therefore, we invite you to submit a revised version of the manuscript that addresses the points raised during the review process. This will require a Major Revision.

We would appreciate receiving your revised manuscript by Jun 28 2020 11:59PM. To enhance the reproducibility of your results, we recommend that if applicable you deposit your laboratory protocols in protocols.io, where a protocol can be assigned its own identifier (DOI) such that it can be cited independently in the future. For instructions see: http://journals.plos.org/plosone/s/submission-guidelines#loc-laboratory-protocols

We look forward to receiving your revised manuscript.

Kind regards,

Andreas Zirlik, MD

Academic Editor

PLOS ONE

Journal Requirements:

'The study was funded by the authors only.'

'None'

'None'

6. Please ensure that you refer to Figure 1 in your text as, if accepted, production will need this reference to link the reader to the figure.

Additional Editor Comments (if provided):

Reviewers' comments:

Reviewer's Responses to Questions

**Comments to the Author**

1. Is the manuscript technically sound, and do the data support the conclusions?

Reviewer #1: Partly

Reviewer #2: Partly

2. Has the statistical analysis been performed appropriately and rigorously? 

Reviewer #1: No

Reviewer #2: I Don't Know

3. Have the authors made all data underlying the findings in their manuscript fully available?

Reviewer #1: No

Reviewer #2: No

4. Is the manuscript presented in an intelligible fashion and written in standard English?

Reviewer #1: Yes

Reviewer #2: Yes

5. Review Comments to the Author

Reviewer #1: Deldar Morad et al. describe the prevalence and concentration of human anti-nucleolus autoantibodies (ANCAb) in a small cohort of individuals with diabetes mellitus type 2 (T2DM), respectively in subjects with microvascular complications (n=38) and without complications (n=43).

Though the topic is interesting per se, there are several severe systematic and design problems.

- Why did the authors choose anti-nucleolus autoantibodies (ANCAb) as their target biomarker? What would be other autoimmune reactions in this group of individuals with T2DM? Picking one autoantibody is not enough to characterize a potential autoimmune component.

- Though the duration of T2DM is described to be similar, the definition of microvascular complications is very vague and several sub-entities are mixed-up in this group.

- There is no clear procedure, how the individuals “without complications” have been examined. Ocular tests only in patients with self-reported complaints is far from being the right screening method in this disease. Therefore, the assignment in the one or the other group might be misleading and explaining the lack of differences.

- Performing one assay of one company in these circumstances is not a suitable approach. Technical problems might have lead to a number of zero values, as shown in Figure 1 – there is no description on assay performance and sensitivity/specificity, neither on false positive or false negative rate.

- The duration of T2DM in both groups (9.0 vs. 5.0 years; P=0.065) is very diverse, as seen in the p value – with 4 years from diagnosis and a potential longer undiagnosed duration before, these groups are not comparable

- As male and female individuals have very different risk factors for vascular diseases and autoimmunity, they should not be mixed up in groups. The more in the small groups in this study.

- The conclusions are not supported by the data and analyses.

Reviewer #2: In this study the authors want to further elucidate the role of autoimmunity in Type 2 diabetes. To do that they investigated whether there are qualitative and quantitative differences in the levels of Human Anti Nucleolus Antibody (ANCAb) in sera of Type 2 diabetes patients with and without diabetes-related complications. The levels of ANCAb were determined by using a ELISA kit for the quantitative determination of ANCAb in serum. The authors did not find differences in reaction to ANCAb or of the amount of ANCAb in the sera of patients without diabetic complication compared to those with complications. There were also no differences in the comparison of specific complications (e.g. neuropathy) with no complications.

I think generally the results could match the criteria for publication but one major issue and some minors should be addressed before:

1) Methods:

The main question for me arises in regard to the performed ELISA and the results obtained by it. In the methods section the authors nicely report the patient selection and clinical evaluation. However the description of the Human Anti Nucleolus Antibody measurement is not sufficient in my point of view:

Please provide detailed protocol of how you performed the assay. Most of the results the authors report are in regard to “reaction to ANCAb”. I can’t find any description of what is actually meant by that or how that measurement was performed, or which cut-offs were used (“In the current study, any reaction of ANCAb with components of the body's healthy cells is was considered a reacted ANCAb.”).

Can you give any references of other studies which used that assay? How are the results when a healthy control group is tested in regard to “any reaction of ANCAb with components of the body healthy cells”.

Minor:

Introduction:

- In their introduction the authors briefly highlight some findings in regard to autoimmunity in diabetes type 1 and 2 and at the end focus on the possible role of anti-nuclear antibodies. Maybe you can provide any thoughts or rationale on why to test the specific subset of Anti Nucleolus Antibody.

- The authors state: “Although different studies suggested that antibodies may be present in the sera of T2DM patients with complications […]”

Please specify.

- Please check the abbreviations. If an abbreviation is introduced it should be used from there on consistently (e.g. T2DM, ANCAb …)

Methods:

- Section “Diagnosis and Measurement Criteria” : Provide units for HbA1c.

- Section“ Human Anti Nucleolus Antibody measurement”:

“Serum ANCAb was performed […]” � Sentence doesn’t make sense, please reframe.

“In the current study, any reaction of ANCAb with components of the body's healthy cells is was considered a reacted ANCAb” � Check again (“is was”)

Results:

- At the end of the manuscript you provide Figure 1 A-D. However the figure is not mentioned in the text at all. Please also provide detailed figure legends.

- In table 5 you provide the number of patients affected by several complications or combinations of complications. However some are missing e.g. the number of patients suffering from CVD-complication is not given and you didn’t test for differences between CVD-complications and without complications. Is there a reason why some of the complications have been left out or have not been tested?

- Paragraph 5 of the result section: “To have a better picture of the reaction to ANCAb in patients with different types of complications, the prevalence of ANCAb reaction in different categories of complications was presented in Table 4”.

Table 4 shows the comparison between controlled and uncontrolled diabetes not the different categories of complications

6. PLOS authors have the option to publish the peer review history of their article (what does this mean?). If published, this will include your full peer review and any attached files.

Reviewer #1: No

Reviewer #2: No

---

## [Author Response · Author response to Decision Letter 0]

29 May 2020

PONE-D-20-08208

Circulating Human Anti Nucleolus Antibody (ANCAb) and Biochemical Parameters In Type 2 Diabetic Patients with and without Complications

PLOS ONE

Dear Mr. Abdulah,

We would appreciate receiving your revised manuscript by Jun 28 2020 11:59PM. To enhance the reproducibility of your results, we recommend that if applicable you deposit your laboratory protocols in protocols.io, where a protocol can be assigned its own identifier (DOI) such that it can be cited independently in the future. For instructions see: http://journals.plos.org/plosone/s/submission-guidelines#loc-laboratory-protocols

• A rebuttal letter that responds to each point raised by the academic editor and reviewer(s). This letter should be uploaded as separate file and labeled 'Response to Reviewers'.

• A marked-up copy of your manuscript that highlights changes made to the original version. This file should be uploaded as separate file and labeled 'Revised Manuscript with Track Changes'.

• An unmarked version of your revised paper without tracked changes. This file should be uploaded as separate file and labeled 'Manuscript'.

Comment: 

Response: The manuscript was send to SCREBENDI Language services. The certificate and its tracked changes was attached as well. 

Comment: 

Response: I responded the question. Please see the application. 

Comment: 

Response: I responded the question. Please see the application. 

Comment: 

Response: There is no restriction to share the data to the public. Just it would be applicable after acceptance. 

'The study was funded by the authors only.'

Comment: 

Response: The study was not funded by any organization. The statement was removed from the manuscript. 

Comment: 

'None'

'None'

Response: I removed the funding statement from the manuscript. 

Comment: 

Response: I filled the form. 

Comment: 

Response: This information was added to the cover letter. Please see it in the attached file. 

Comment: 

6. Please ensure that you refer to Figure 1 in your text as, if accepted, production will need this reference to link the reader to the figure. 

Response: The figures were referenced to the text in the results. Please see the manuscript. 

Comment: 

Additional Editor Comments (if provided): 

Response: We have not any supporting file for this manuscript. 

Reviewers' comments:

Reviewer's Responses to Questions

Comments to the Author

Comment: 

1. Is the manuscript technically sound, and do the data support the conclusions?

Reviewer #1: Partly

Reviewer #2: Partly

Response: We tried to make a connection between the results and conclusions. Please see the manuscript. 

 Comment: 

2. Has the statistical analysis been performed appropriately and rigorously? 

Reviewer #1: No

Reviewer #2: I Don't Know

Response: I checked the statistical extractions of the manuscript. Please see it. 

Comment: 

3. Have the authors made all data underlying the findings in their manuscript fully available?

Reviewer #1: No

Reviewer #2: No

Response: We uploaded the SPSS file for the review process if required. 

 Comment: 

4. Is the manuscript presented in an intelligible fashion and written in standard English?

Reviewer #1: Yes

Reviewer #2: Yes

Response: Thanks. 

 Comment: 

5. Review Comments to the Author 

Reviewer #1: Deldar Morad et al. describe the prevalence and concentration of human anti-nucleolus autoantibodies (ANCAb) in a small cohort of individuals with diabetes mellitus type 2 (T2DM), respectively in subjects with microvascular complications (n=38) and without complications (n=43).

Though the topic is interesting per se, there are several severe systematic and design problems.

 Comment: 

- Why did the authors choose anti-nucleolus autoantibodies (ANCAb) as their target biomarker? 

Response: The role of Anti-nuclear antibodies (ANA) has been investigated in few studies with controversial findings. We hypothesized that the autoimmunity could be developed within an anti-nucleolus rather than nuclear level. Therefore, we measured anti-nucleolus autoantibodies (ANCAb) between the patients with and without micro-complications in T2DM patients. 

 Comment: 

What would be other autoimmune reactions in this group of individuals with T2DM? Picking one autoantibody is not enough to characterize a potential autoimmune component. 

Response: It is right, but we had not sufficient financial resources to include other autoantibodies in the study. 

 Comment: 

- Though the duration of T2DM is described to be similar, the definition of microvascular complications is very vague and several sub-entities are mixed-up in this group.

Response: The primary endpoint was the first record of one of the following 12 cardiovascular presentations in any of the

data sources: stable angina, unstable angina, myocardial infarction, unheralded coronary death, heart failure, transient ischaemic attack, ischaemic stroke, subarachnoid haemorrhage, intracerebral haemorrhage, peripheral arterial disease, abdominal aortic aneurysm, and a composite outcome consisted of cardioversion, ventricular arrhythmia, implantable cardioverter defibrillator, cardiac arrest, or sudden cardiac death (Reference: Shah, A.D., Langenberg, C., Rapsomaniki, E., Denaxas, S., Pujades-Rodriguez, M., Gale, C.P., Deanfield, J., Smeeth, L., Timmis, A. and Hemingway, H., 2015. Type 2 diabetes and incidence of cardiovascular diseases: a cohort study in 1· 9 million people. The lancet Diabetes & endocrinology, 3(2), pp.105-113.

Comment: 

- There is no clear procedure, how the individuals “without complications” have been examined. 

Response: The patients who attended the out-patient clinic for medical check-up or receiving their regular medications were screened physically and clinically by the study internist. The study internist has weekly regular clinical duty in the out-patient clinic. The patients were screened for the possible complications of T2DM and consisted microvascular diseases, neuropathy, nephropathy, and retinopathy. The patients who were diagnosed to have one of the mentioned above complications following diabetes were categorized as T2DM patients with complication. The T2DM patients who did not have the complications were categorized as T2DM patients without complications. Please see the manuscript. 

 Comment: 

Ocular tests only in patients with self-reported complaints is far from being the right screening method in this disease. Therefore, the assignment in the one or the other group might be misleading and explaining the lack of differences.

Response: We did not refer all patients to an ophthalmologist. We felt that it is not necessary to refer all patients to the ophthalmologist. Therefore, only patients who had a complaint on their visions during investigation were referred to the eye clinic. 

 Comment: 

- Performing one assay of one company in these circumstances is not a suitable approach. 

Response: We had limited financial resources to purchase the material assays of another company as well, since the study was not funded by any organization. 

 Comment: 

Technical problems might have led to a number of zero values, as shown in Figure 1 – there is no description on assay performance and sensitivity/specificity, neither on false positive or false negative rate.

Response: We added the required information. Please see the manuscript. 

 Comment: 

- The duration of T2DM in both groups (9.0 vs. 5.0 years; P=0.065) is very diverse, as seen in the p value – with 4 years from diagnosis and a potential longer undiagnosed duration before, these groups are not comparable

Response: I checked the diverse of the disease duration in a dot plot. They are not diverse. They have just some outliers in both groups. But, the outliers are not extreme. Please see the following graph.

 Comment: 

- As male and female individuals have very different risk factors for vascular diseases and autoimmunity, they should not be mixed up in groups. The more in the small groups in this study.

Response: Both groups have more females than males with the statistically comparable way (P=0.244). We did the comparison of ANCAb between males and females in Table 3. However, we could not include only males or females, because we could not obtain the target number of complicated patients within the planned time period. In that case, we needed longer time period to include the target number of patients into the study. 

 Comment:

- The conclusions are not supported by the data and analyses.

Response: I did revision in the conclusions. Please see the manuscript. 

Reviewer #2: In this study the authors want to further elucidate the role of autoimmunity in Type 2 diabetes. To do that they investigated whether there are qualitative and quantitative differences in the levels of Human Anti Nucleolus Antibody (ANCAb) in sera of Type 2 diabetes patients with and without diabetes-related complications. The levels of ANCAb were determined by using a ELISA kit for the quantitative determination of ANCAb in serum. The authors did not find differences in reaction to ANCAb or of the amount of ANCAb in the sera of patients without diabetic complication compared to those with complications. There were also no differences in the comparison of specific complications (e.g. neuropathy) with no complications.

I think generally the results could match the criteria for publication but one major issue and some minors should be addressed before:

 Comment: 

1) Methods:

The main question for me arises in regard to the performed ELISA and the results obtained by it. In the methods section the authors nicely report the patient selection and clinical evaluation. However the description of the Human Anti Nucleolus Antibody measurement is not sufficient in my point of view: 

Please provide detailed protocol of how you performed the assay. Most of the results the authors report are in regard to “reaction to ANCAb”. I can’t find any description of what is actually meant by that or how that measurement was performed, or which cut-offs were used (“In the current study, any reaction of ANCAb with components of the body's healthy cells is was considered a reacted ANCAb.”). 

Response: The required information was added to the methods section. Please see the manuscript. 

 Comment: 

Can you give any references of other studies which used that assay? How are the results when a healthy control group is tested in regard to “any reaction of ANCAb with components of the body healthy cells”. 

Response: With respect to the ANCAb, this is the first study that used ANCAb for the T2DM patients. However, about ANA, there are few studies; for example the following:

Grainger, D.J. and Bethell, H.W.L., 2002. High titres of serum antinuclear antibodies, mostly directed against nucleolar antigens, are associated with the presence of coronary atherosclerosis. Annals of the rheumatic diseases, 61(2), pp.110-114.

Janahi, N.M., Santos, D., Blyth, C., Bakhiet, M. and Ellis, M., 2015. Diabetic peripheral neuropathy, is it an autoimmune disease?. Immunology letters, 168(1), pp.73-79.

Comment: 

Minor:

Introduction:

- In their introduction the authors briefly highlight some findings in regard to autoimmunity in diabetes type 1 and 2 and at the end focus on the possible role of anti-nuclear antibodies. Maybe you can provide any thoughts or rationale on why to test the specific subset of Anti Nucleolus Antibody. 

Response: We found few investigation that examined the role of antinuclear antibodies (ANA) in development of complications in T2DM patients. We hypothesized that development of complications in T2DM patients may be searched in an anti- nucleolus level rather than nuclear one. Therefore, we decided to work on the anti- nucleolus layer rather than nuclear one. 

 Comment: 

- The authors state: “Although different studies suggested that antibodies may be present in the sera of T2DM patients with complications […]”

Please specify.

Response: We specified the studies. Please see the manuscript. 

 Comment: 

- Please check the abbreviations. If an abbreviation is introduced it should be used from there on consistently (e.g. T2DM, ANCAb …)

Response: I checked and revised as appropriate. 

 Comment: 

Methods:

- Section “Diagnosis and Measurement Criteria” : Provide units for HbA1c. 

Response: It was provided. Please see the manuscript. 

 Comment: 

- Section“ Human Anti Nucleolus Antibody measurement”:

“Serum ANCAb was performed […]” � Sentence doesn’t make sense, please reframe.

“In the current study, any reaction of ANCAb with components of the body's healthy cells is was considered a reacted ANCAb” � Check again (“is was”)

Response: They were revised as appropriate. Please see the manuscript.

 Comment: 

Results:

- At the end of the manuscript you provide Figure 1 A-D. However the figure is not mentioned in the text at all. Please also provide detailed figure legends.

Response: That is right, I did the reference to the text. Please see the manuscript.

 Comment: 

- In table 5 you provide the number of patients affected by several complications or combinations of complications. However some are missing e.g. the number of patients suffering from CVD-complication is not given and you didn’t test for differences between CVD-complications and without complications. Is there a reason why some of the complications have been left out or have not been tested?

Response: We had not sufficient number of patients with CVD in both group to make a comparison. 

 Comment: 

- Paragraph 5 of the result section: “To have a better picture of the reaction to ANCAb in patients with different types of complications, the prevalence of ANCAb reaction in different categories of complications was presented in Table 4”.

Table 4 shows the comparison between controlled and uncontrolled diabetes not the different categories of complications

 Response: It was changed to the Table 5. Please see the manuscript. 

 6. PLOS authors have the option to publish the peer review history of their article (what does this mean?). If published, this will include your full peer review and any attached files.

Do you want your identity to be public for this peer review? For information about this choice, including consent withdrawal, please see our Privacy Policy.

Reviewer #1: No

Reviewer #2: No

---

## [Decision Letter · Decision Letter 1]

17 Jun 2020

PONE-D-20-08208R1

Circulating Human Anti Nucleolus Antibody (ANCAb) and Biochemical Parameters In Type 2 Diabetic Patients with and without Complications

PLOS ONE

Dear Dr. Abdulah,

Thank you for submitting your manuscript to PLOS ONE. After careful consideration, we feel that it has merit but does not fully meet PLOS ONE’s publication criteria as it currently stands. Therefore, we invite you to submit a revised version of the manuscript that addresses the points raised during the review process. Please address all points of Reviewer 2.

We look forward to receiving your revised manuscript.

Kind regards,

Andreas Zirlik, MD

Academic Editor

PLOS ONE

Reviewers' comments:

Reviewer's Responses to Questions

**Comments to the Author**

1. If the authors have adequately addressed your comments raised in a previous round of review and you feel that this manuscript is now acceptable for publication, you may indicate that here to bypass the “Comments to the Author” section, enter your conflict of interest statement in the “Confidential to Editor” section, and submit your "Accept" recommendation.

Reviewer #1: (No Response)

Reviewer #2: (No Response)

2. Is the manuscript technically sound, and do the data support the conclusions?

Reviewer #1: Partly

Reviewer #2: Partly

3. Has the statistical analysis been performed appropriately and rigorously? 

Reviewer #1: N/A

Reviewer #2: I Don't Know

4. Have the authors made all data underlying the findings in their manuscript fully available?

Reviewer #1: Yes

Reviewer #2: Yes

5. Is the manuscript presented in an intelligible fashion and written in standard English?

Reviewer #1: Yes

Reviewer #2: Yes

6. Review Comments to the Author

Reviewer #1: Some of the issues have been addressed, but several important points, like the ocular tests in ALL patients (which is mandatory for every DM subject and would only support the statistical outcome in terms of microvascular complications) seem to be not available.

In addition, the assay data are not convincing and not supportive for the author's conclusions.

Priority for publication is therefore low.

Reviewer #2: In my opinion, the revised versionn of the manuscript provided by the authors has improved in quality but there are still some issues that need to be addressed:

1) Can you give any references of other studies which used that assay? How are the results when a healthy control group is tested in regard to “any reaction of ANCAb with components of the body healthy cells”.

Response: With respect to the ANCAb, this is the first study that used ANCAb for the T2DM patients. However, about ANA, there are few studies; for example the following: Grainger, D.J. and Bethell, H.W.L., 2002. High titres of serum antinuclear antibodies, mostly directed against nucleolar antigens, are associated with the presence of coronary atherosclerosis. Annals of the rheumatic diseases, 61(2), pp.110-114.

Janahi, N.M., Santos, D., Blyth, C., Bakhiet, M. and Ellis, M., 2015. Diabetic peripheral neuropathy, is it an autoimmune disease?. Immunology letters, 168(1), pp.73-79.

Comment: I understand this is the first study which tested for ANCAb in T2DM patients. My question was if there are data on ANCAb reaction in healthy controls. In their discussion the authors attribute the reported reactions to ANCAb to latent autoimmune diabetes in adults. I don't think this conclusion is valid if there is no comparison to a group of matched healthy controls. I would suggest to include a control group of matched healthy controls to see if there is a role of ANCAb reaction in T2DM patients at all.

2) One of the differences of the two groups was waist circumference with higher values in the non-complicated group (107.95 to 102.30). Also there was a trend to higher BMI in the non-complicated group. Since adipose tissue is known to to contribute to secretion of autoimmune antibodies that potentially could disguise an estimated higher amount of ANCAb reaction in more complicated disease. Maybe the authors could add that in their discussion.

3) I think many of the points reviewer 1 mentioned are valid (like verification by another method or looking for different autoantibodies) and if they can not be resolved they should at least be discussed or added in the limitation section.

7. PLOS authors have the option to publish the peer review history of their article (what does this mean?). If published, this will include your full peer review and any attached files.

Reviewer #1: No

Reviewer #2: No

---

## [Author Response · Author response to Decision Letter 1]

22 Jun 2020

PONE-D-20-08208R1

Circulating Human Anti Nucleolus Antibody (ANCAb) and Biochemical Parameters In Type 2 Diabetic Patients with and without Complications

PLOS ONE

Dear Dr. Abdulah,

Kind regards,

Andreas Zirlik, MD

Academic Editor

PLOS ONE

6. Review Comments to the Author

Comment: Reviewer #1: Some of the issues have been addressed, but several important points, like the ocular tests in ALL patients (which is mandatory for every DM subject and would only support the statistical outcome in terms of microvascular complications) seem to be not available. 

In addition, the assay data are not convincing and not supportive for the author's conclusions.

Priority for publication is therefore low.

Response: As we mentioned previously, we did not perform the ocular tests for all patients. The patients were physically and clinically screened by the study internist for any possible complications. The patients who had vision complaints were referred to an ophthalmologist only. We honestly report that the ocular tests were not performed for all patients. We added into the limitations of the study as well. 

Reviewer #2: In my opinion, the revised version of the manuscript provided by the authors has improved in quality but there are still some issues that need to be addressed:

Comment: 1) Can you give any references of other studies which used that assay? How are the results when a healthy control group is tested in regard to “any reaction of ANCAb with components of the body healthy cells”. 

Response: We did not find other references that used this test. But, the following is the link of the test company with required information. We attached the catalogue of the company for this test into the application. 

The link of the test of the company: 

https://www.mybiosource.com/human-elisa-kits/anti-nucleolus-antibody-ana/7228076

Also, we found this test from another company as well. The following is the link of this test for the Human anti-nucleolus antibody and Mouse anti-nucleolus antibody test, respectively.

https://www.biocompare.com/9956-Assay-Kit/2796027-Human-anti-nucleolus-antibody-ANA-ELISA-Kit/?pda=9956|2796027_0_0||11|ANA 

https://www.biocompare.com/25138-Assay-Kit/13280700-Mouse-anti-nucleolus-antibody-ANA-ELISA-Kit/?pda=9956|13280700_0_0||27|ANA 

Concerning healthy controls, it was good to include the healthy controls in the study. However, we had financial restrictions for this study. Therefore, we added as a limitation into the study. 

Comment: 

Response: With respect to the ANCAb, this is the first study that used ANCAb for the T2DM patients. However, about ANA, there are few studies; for example the following: Grainger, D.J. and Bethell, H.W.L., 2002. High titres of serum antinuclear antibodies, mostly directed against nucleolar antigens, are associated with the presence of coronary atherosclerosis. Annals of the rheumatic diseases, 61(2), pp.110-114.

Janahi, N.M., Santos, D., Blyth, C., Bakhiet, M. and Ellis, M., 2015. Diabetic peripheral neuropathy, is it an autoimmune disease?. Immunology letters, 168(1), pp.73-79.

Comment: I understand this is the first study which tested for ANCAb in T2DM patients. My question was if there are data on ANCAb reaction in healthy controls. In their discussion the authors attribute the reported reactions to ANCAb to latent autoimmune diabetes in adults. I don't think this conclusion is valid if there is no comparison to a group of matched healthy controls. I would suggest to include a control group of matched healthy controls to see if there is a role of ANCAb reaction in T2DM patients at all.

Response: We could not include the healthy controls in this study due to financial resources. If we included the healthy controls, the sample size would be reduced more than this. We added this point to the study limitations as well. Please see the manuscript. 

 Comment: 2) One of the differences of the two groups was waist circumference with higher values in the non-complicated group (107.95 to 102.30). Also there was a trend to higher BMI in the non-complicated group. Since adipose tissue is known to contribute to secretion of autoimmune antibodies that potentially could disguise an estimated higher amount of ANCAb reaction in more complicated disease. Maybe the authors could add that in their discussion. 

Response: We added the relevant information about this point. Please see the manuscript. 

Comment: 

3) I think many of the points reviewer 1 mentioned are valid (like verification by another method or looking for different autoantibodies) and if they cannot be resolved they should at least be discussed or added in the limitation section. 

Response: We added this point to the limitation. Please see the manuscript. 

Comment: 

7. PLOS authors have the option to publish the peer review history of their article (what does this mean?). If published, this will include your full peer review and any attached files.

Response: It is OK. 

---

## [Editor Report · Decision Letter 2]

21 Jul 2020

Circulating Human Anti Nucleolus Antibody (ANCAb) and Biochemical Parameters In Type 2 Diabetic Patients with and without Complications

PONE-D-20-08208R2

Dear Dr. Abdulah,

We’re pleased to inform you that your manuscript has been judged scientifically suitable for publication and will be formally accepted for publication once it meets all outstanding technical requirements.

Kind regards,

Andreas Zirlik, MD

Academic Editor

PLOS ONE
---

## [Editor Report · Acceptance letter]

3 Aug 2020

PONE-D-20-08208R2 

Circulating Human Anti Nucleolus Antibody (ANCAb) and Biochemical Parameters In Type 2 Diabetic Patients with and without Complications 

Dear Dr. Abdulah:

I'm pleased to inform you that your manuscript has been deemed suitable for publication in PLOS ONE. Congratulations! Your manuscript is now with our production department. 

Kind regards, 

on behalf of

Univ. Prof. Dr. Andreas Zirlik 

Academic Editor

PLOS ONE